# Elimination of Lumbar Plexus Injury by Changing the Entry Point and Traction Direction of the Psoas Major Muscle in Transpsoas Lateral Lumbar Spine Surgery

**DOI:** 10.3390/medicina59040730

**Published:** 2023-04-08

**Authors:** Hidetoshi Nojiri, Takatoshi Okuda, Hiromitsu Takano, Motoshi Gomi, Ryosuke Takahashi, Arihisa Shimura, Shota Tamagawa, Takeshi Hara, Yukoh Ohara, Muneaki Ishijima

**Affiliations:** 1Department of Orthopedic Surgery, Juntendo University, Tokyo 113-8421, Japan; 2Spine and Spinal Cord Center, Juntendo University Hospital, Tokyo 113-8421, Japan; 3Department of Neurosurgery, Juntendo University, Tokyo 113-8421, Japan

**Keywords:** lumbar plexus injury, lateral lumbar interbody fusion, transpsoas approach, anterior psoas splitting approach, neuromonitoring, neurological complication

## Abstract

*Background and Objectives:* The lateral approach is commonly used for anterior column reconstruction, indirect decompression, and fusion in patients with lumbar degenerative diseases and spinal deformities. However, intraoperative lumbar plexus injury may occur. This is a retrospective comparative study to investigate and compare neurological complications between the conventional lateral approach and a modified lateral approach at L4/5. *Materials and Methods:* Patients with a lumbar degenerative disease requiring single-level intervertebral fusion at L4/5 were included and categorized into group X and group A. Patients in group X underwent conventional extreme lateral interbody fusion, while those in group A underwent a modified surgical procedure that included splitting of the anterior third of the psoas muscle, which was dilated by the retractor on the anterior third of the intervertebral disc. The incidence of lumbar plexus injury, defined as a decrease of ≥1 grade on manual muscle testing of hip flexors and knee extensors and sensory impairment of the thigh for ≥3 weeks, on the approach side, was investigated. *Results:* Each group comprised 50 patients. No significant between-group differences in age, sex, body mass index, and approach side were observed. There was a significant between-group difference in intraoperative neuromonitoring stimulation value (13.1 ± 5.4 mA in group X vs. 18.5 ± 2.3 mA in group A, *p* < 0.001). The incidence of neurological complications was significantly higher in group X than in group A (10.0% vs. 0.0%, respectively, *p* < 0.05). *Conclusions:* In our modified procedure, the anterior third of the psoas muscle was entered and split, and the intervertebral disc could be reached without damaging the lumbar plexus. When performing lumbar surgery using the lateral approach, lumbar plexus injury can be avoided by following surgical indication criteria based on the location of the lumbar plexus with respect to the psoas muscle and changing the transpsoas approach to the intervertebral disc.

## 1. Introduction

The lateral approach is used in lumbar surgery for the surgical treatment of lumbar degenerative diseases and spinal deformities. It has the advantage of allowing effective anterior column reconstruction and decompression fixation with minimal invasiveness. Because the lumbar discs and vertebrae are approached from the lateral side, there are various techniques for handling the psoas major muscle, which is located on the lateral side of the lumbar spine, and the position and direction of entry into the psoas major muscle must be carefully considered. Oblique lateral intervertebral fusion (OLIF) or anterior to psoas (ATP) approaches essentially enter from the anterior edge of the psoas major muscle and pull the muscle posteriorly [1,2]. In contrast, direct lateral or extreme lateral interbody fusion (XLIF) splits and enters the psoas major muscle and pulls the muscle anteriorly [3]. Both approaches have specific complications, resulting from the lumbar lateral approach, such as retroperitoneal organ injury and nerve injury in the psoas major muscle [4,5]. Damage to the lumbar plexus on the approach side occurs with either method, leading to motor and sensory impairment in the lower limbs, and postoperative recovery is significantly delayed, resulting in significantly reduced patient satisfaction with surgery. Therefore, measures for avoiding complications are necessary [6,7,8]. The incidence of complications is particularly high when the transpsoas approach is used at L4/5 [9,10], and avoiding neurological complications at this level is challenging. On the other hand, Ohiorhenuan IE, et al. described that the transpsoas lateral lumbar interbody fusion at L4/5 is safe, with careful use of intraoperative neuromonitoring and judicious retraction, even for patients with the forward shift of the psoas muscle [11]. Our previous cadaver study that investigated the location of the lumbar plexus relative to the psoas muscle found that the lumbar plexus ran alongside the posterior third of the psoas muscle irrespective of the muscle shape and position [12]. The findings of that study inspired us to develop indication criteria for the use of the lateral approach in lumbar spine surgery using preoperative magnetic resonance imaging (MRI) in the axial view. We also modified the conventional lateral approach with respect to the position of entry into the psoas major muscle and the direction of muscle traction. We compared the occurrence of neurological complications between the conventional approach and our modified approach in the present study.

## 2. Materials and Methods

Patients with a lumbar degenerative disease with neurological symptoms requiring single-level interbody fusion at L4/5 were included in this study. Patients with anatomical anomalies of the iliac crest and major vessels were excluded to ensure a safe approach. To evaluate the position of the psoas muscle relative to the intervertebral disc, the relative position (RP) value was measured using T2-weighted MRI in the axial view (Figure 1A). The RP value is the position of the posterior third of the psoas muscle relative to the intervertebral disc in the anterior-posterior plane, with 0% and 100%, indicating the posterior and anterior edges of the intervertebral disc, respectively. A RP value of ≤66.7%, which indicated that the posterior third of the psoas muscle was situated behind the anterior third of the intervertebral disc, was considered an indication for using the lateral approach (Figure 1B,C). Patients were classified into group X or group A based on the procedure they underwent. Since the procedure was changed in 2018, patients operated on before the change are in group X, and patients operated on after the change are in group A. In group X (Figure 2A), patients underwent XLIF, using the conventional procedure [3], in which the lateral side of the psoas muscle was exposed, the position of the intervertebral disc was confirmed using fluoroscopy, and the split psoas muscle was pulled forward after driving a shim into the intervertebral disc, as per the instructions for XLIF by Nuvasive, Inc.(San Diego, CA, USA) (Figure 3A,B). In group A (Figure 2B), we used the anterior psoas splitting approach, a modified technique that included splitting of the anterior third of the psoas muscle (Figure 3C). Under fluoroscopic guidance, the anterior third of the intervertebral disc was reached, and the psoas muscle was pulled posteriorly using Crestline, manufactured by Nuvasive, Inc. (Figure 3D). With both methods, intraoperative neuromonitoring was performed, and the minimum stimulation value (in mA) was measured using the NVM5 (Nuvasive, Inc., San Diego, CA, USA). A green signal (≥11 mA) on the NVM5 monitor indicated that there were no nerves near the stimulator tip placed in or under the split muscle, whereas a yellow (6–10 mA) or red (≤5 mA) signal was considered a warning that the stimulator tip was near a nerve. All surgeries were performed by the same experienced surgeon at two university hospitals, and all patients were followed for at least two years. A decrease of ≥1 grade in manual muscle testing of hip flexors and knee extensors on the approach side and sensory impairment of the thigh for ≥3 weeks were considered neurological complications. Sensory impairment localized to the circumference of the wound or inguinal region and abdominal muscle paralysis was not considered neurological complications in this study. Age, sex, height, body weight, body mass index (BMI), approach side, the position of the psoas muscle relative to the disc, duration of surgery, estimated blood loss, the number of warning signals on the NMV5, and incidence of neurological complications were investigated in each group. The *t*-test and the Mann–Whitney U test were used for between-group comparisons, and the significance level was set at 5% on both sides. This study has been approved by the research ethics committee of Juntendo University (Approval code: 20-359, approved on 5 February 2021). Informed consent was obtained from all the participants in this study.

## 3. Results

Fifty patients in each group had lumbar spondylolisthesis with instability, lumbar spinal canal stenosis, lumbar foraminal stenosis, or degenerative disc disease, and they underwent single-level lateral lumbar interbody fusion at L4/5.

The mean age (years) was 71.8 ± 10.2 in group X and 70.4 ± 12.1 in group A, showing no significant difference. In group X and group A, 26.0% and 20.0% of patients were men, respectively. The mean BMI was 25.6 ± 3.8 in group X and 24.1 ± 3.1 in group A. A left-sided approach was used in 47 and 48 patients in group X and group A, respectively. The RP value (%) was 38.8 ± 16.2 in group X and 42.1 ± 20.1 in group A, and no significant difference was observed between the two groups (Table 1). The duration of surgery (min) was 142.3 ± 30.8 in group X and 133.7 ± 25.9 in group A, and the estimated blood loss (g) was 29.9 ± 18.9 in group X and 27.0 ± 20.2 in group A, with no significant differences between the groups (Table 2). There was a significant difference in the average intraoperative neuromonitoring stimulus value between the two groups (13.1 ± 5.4 mA in group X and 18.5 ± 2.3 mA in group A, *p* < 0.001, Table 2, Figure 4). Yellow or red signals were observed on intraoperative nerve monitoring in 20 patients (40.0%) in group X, but these signals were not observed in any patient in group A (*p* < 0.001, Table 2, Figure 4). Five patients (10.0%) in group X experienced neurological complications, while no patient in group A experienced neurological complications (*p* < 0.05, Table 2, Figure 4). The five patients with neuropathy had significantly lower neuromonitoring stimulation values than those without (5.8 ± 2.4 mA vs. 16.3 ± 4.4 mA, *p* < 0.001, Figure 4). All patients in group X who had neuropathy developed both motor and sensory deficits, four improved within three months, but one was to remain permanently. There were no cases of major vascular, intestinal, or ureteral injuries in either group.

## 4. Discussion

### 4.1. Indication for Surgery and Patient Selection

The lateral approach has not been found useful for lumbar interbody fusion because it causes more complications than the posterior approach [13]. A previous report regarded the forward shift of the psoas major muscle (rising psoas sign) as a risk factor for the occurrence of neurological complications in patients undergoing XLIF [14]. Even recently, several articles have investigated the risk factors for nerve injury and the safety zone for avoiding nerve injury in transpsoas lateral lumbar interbody fusion at L4/5 [15,16]. Our previous study investigated the location of the lumbar plexus in 27 cadavers to determine factors that could prevent neurological complications and concluded that the lumbar plexus was located at the posterior third of the psoas muscle, regardless of the shape and height of the psoas muscle [12]. Based on these findings, it is expected that forward shift of the psoas major muscle will result in the posterior third of the psoas major muscle being located at the intervertebral disc entry, and the degree of forward shift could aid in patient selection. The current study did not include patients with the posterior third of the psoas major muscle anterior to the anterior third of the intervertebral disc (RP value ≥ 66.7%), but it failed to completely prevent nerve complications. RP values were found to reduce the risk of nerve injury, but not to eradicate it on their own.

### 4.2. Direct and Indirect Nerve Damage

According to a study that compared the incidence of neural complications between XLIF and OLIF, the incidence of neural complications was significantly higher with XLIF than with OLIF. However, since OLIF is also associated with neural complications, nerve injury can occur without direct nerve damage [4]. Therefore, lumbar plexus injury in patients undergoing lumbar surgery with the lateral approach may be due to direct or indirect nerve injury caused by strong pressure on the psoas muscle. Several studies have also reported that nerve damage is associated with the time for which the psoas muscle is under traction and the duration of surgery [17,18]. The more anteriorly the psoas muscles are situated, the higher the risk of direct injury, and the harder and longer the psoas muscles are pressed, the higher the risk of indirect injury. To avoid causing either injury, the split must be made in a location that is not close to the nerve and does not apply a strong force to the muscle.

### 4.3. Surgical Indication Criteria and Modified Surgical Approach

The use of an L4/5 approach route has been debated. In this study, we established indication criteria for avoiding nerve damage based on the relative positional relationship between the psoas muscle and the intervertebral disc on axial-view MRI. Furthermore, by properly selecting the point of entry and direction of approach to the psoas muscle, it is possible to reach the intervertebral disc, minimally expose the intervertebral disc, and gently pull the psoas muscle without damaging the lumbar plexus. In this study, we used the transpsoas approach by selecting patients based on accurately set indication criteria and using a different path into the psoas muscle. By considering the relative position of the psoas muscle and the intervertebral disc, and by changing the way the psoas muscle is entered and split, it is now possible to perform lateral lumbar interbody fusion without lumbar plexus injury. There was a significant difference in the intraoperative neuromonitoring stimulation value between the two groups, suggesting that the distance from the lumbar plexus to the point at which the muscle was split differed between the groups. This finding provides electrophysiological evidence that the lumbar spine can be operated at a sufficient distance from the lumbar plexus even at the L4/5 level. This modified splitting approach has proven to cause no femoral nerve damage at all by operating at the proper distance from the nerve. Neuromonitoring is essential for this procedure to detect the distance to the nerve and prevent nerve damage, as shown in the previous literature [19,20].

There are several limitations. First, this is a retrospective study with a small number of subjects with less significant anterior shift of the psoas muscle. Second, the maximum value that could be measured for the stimulus threshold was 20 mA, so it was not possible to accurately measure more than that, and the distance was not accurately measured. Finally, while we evaluated the motor function of the lumbar plexus and damage to the femoral nerve, we did not evaluate sensory branches and superficial cutaneous nerves, such as the genitofemoral nerve, iliohypogastric nerve, and ilioinguinal nerve. Although sensory impairment at the circumference of the wound and inguinal region does not cause significant harm, the incidence is not low [21]. Therefore, we would prefer to eliminate these complications in the future. Further improvements in the procedure are required.

## 5. Conclusions

Lumbar plexus injury during lateral lumbar interbody fusion at L4/5 can be prevented by establishing surgical indication criteria based on the location of the lumbar plexus and psoas muscle, as well as by changing the surgical approach to the psoas muscle. We reported the indication criteria and a modified surgical approach that enabled us to perform a transpsoas lateral approach in lumbar surgery without neurological complications.

## Figures and Tables

**Figure 1 medicina-59-00730-f001:**
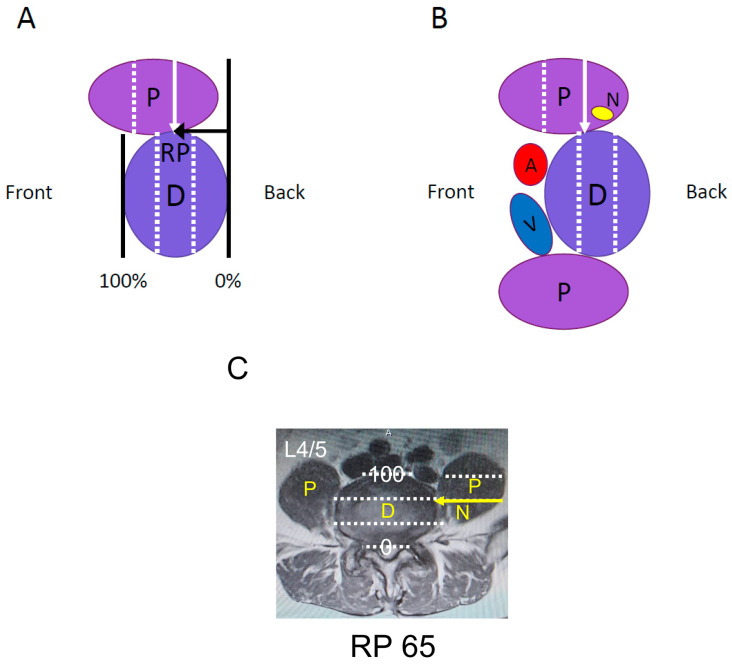
(**A**) Schema, showing the technique for measuring the relative position (RP) value on an axial view of a T2-weighted magnetic resonance image. The RP value was based on the position of the posterior third of the psoas muscle relative to the intervertebral disc in the anterior-posterior plane, with 0% and 100%, indicating the posterior and anterior edges of the intervertebral disc, respectively. (**B**) A RP value of ≤66.7% was considered an indication for lateral approach-interbody fusion. (**C**) This patient is indicated for transpsoas lateral approach with a RP value of 65 on MRI evaluation. White dotted lines indicate the trisecting lines of the intervertebral disc and psoas muscle. Yellow arrow indicates the posterior 1/3 line of the psoas muscle. P: psoas muscle, D: intervertebral disc, A: aorta, V: vein, N: nerve.

**Figure 2 medicina-59-00730-f002:**
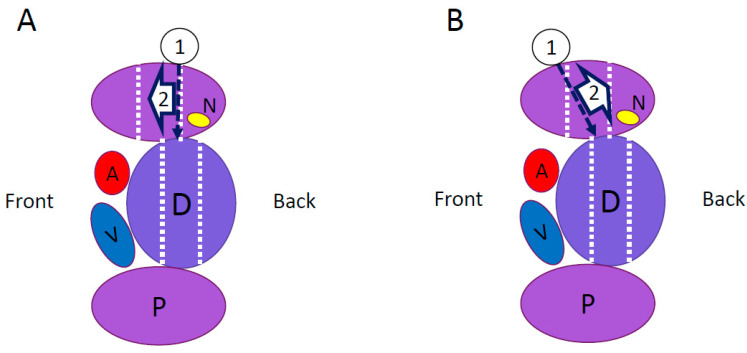
(**A**) In group X, after exposing the lateral aspect of the psoas muscle, the position of the intervertebral disc was confirmed by using fluoroscopy. To reach the center of the intervertebral disc, the posterior third of the psoas muscle was pulled forward with a shim. (**B**) In group A, a different method was used for approaching the psoas muscle and retracting its anterior third. Under fluoroscopic guidance, the anterior third of the intervertebral disc was reached, and the psoas muscle was pulled posteriorly. ➀: Entry point of the psoas muscle, ➁: direction of retractor opening, P: psoas muscle, D: intervertebral disc, A: aorta, V: vein, and N: nerve.

**Figure 3 medicina-59-00730-f003:**
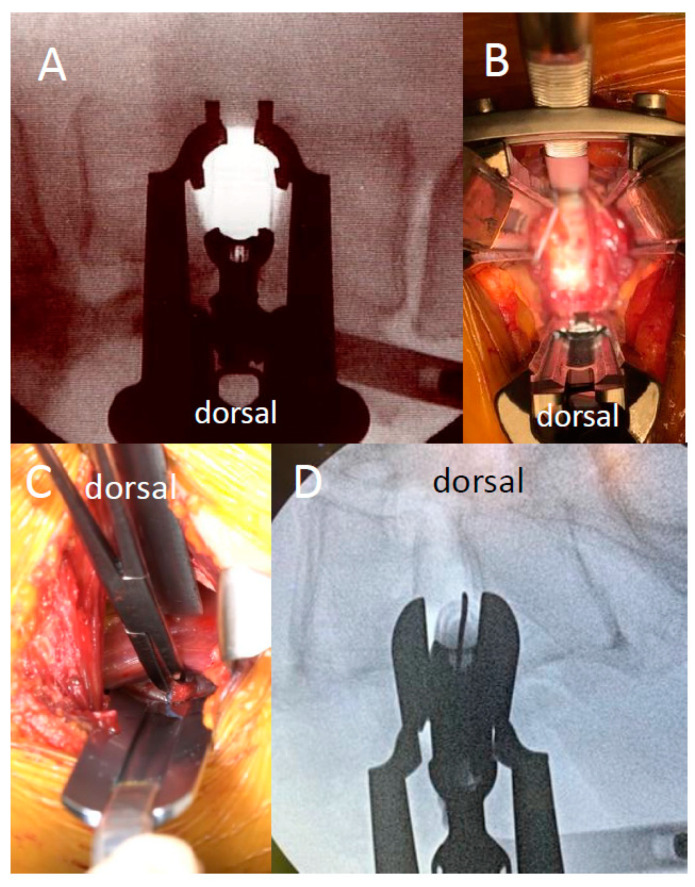
(**A**) In group X, after exposing the lateral portion of the psoas major muscle, we split and entered it with fluoroscopic view of the intervertebral disc’s location. (**B**) In group X, the split psoas muscle was pulled forward after driving a shim into the intervertebral disc. (**C**) In group A, the anterior third of the psoas muscle was gently split, and the intervertebral disc was exposed. (**D**) In group A, the anterior third of the intervertebral disc was reached under fluoroscopy, and the psoas muscle was dilated with a retractor.

**Figure 4 medicina-59-00730-f004:**
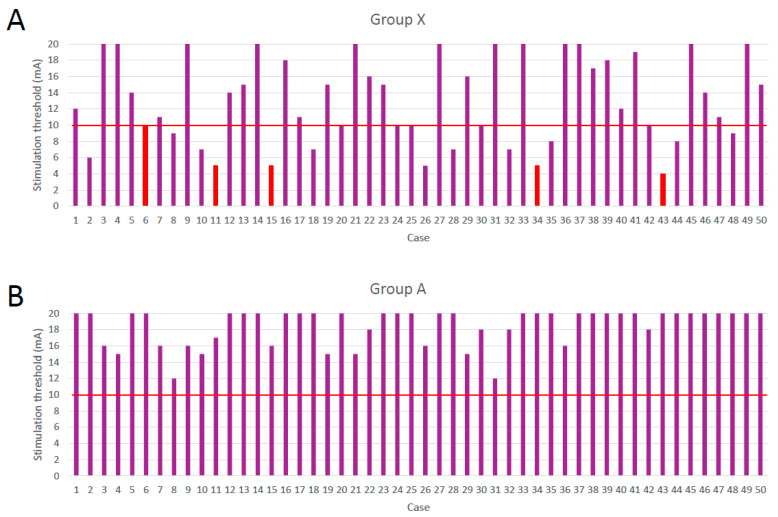
In (**A**) Group X and (**B**) group A, the stimulation value is measured during intraoperative neuromonitoring using the NVM5. Yellow (6–10 mA) or red (≤5 mA) signals were considered warning signals. Patients 6, 11, 15, 34, and 43 in group X experienced neurological complications.

**Table 1 medicina-59-00730-t001:** Preoperative demographic characteristics of the patients in each group.

	Group X (*n* = 50)	Group A (*n* = 50)	*p*-Value
Age (years)	71.8 ± 10.2	70.4 ± 12.1	N.S.
Sex (male/female)	M13/F37	M10/F40	N.S.
Body mass index	25.6 ± 3.8	24.3 ± 3.1	N.S.
Approach (right/left)	R3/L47	R2/L48	N.S.
Relative position value (%)	38.8 ± 16.2	42.1 ± 20.1	N.S.

N.S.: not significant.

**Table 2 medicina-59-00730-t002:** Perioperative and postoperative findings of the patients in each group.

	Group X (*n* = 50)	Group A (*n* = 50)	*p*-Value
Estimated blood loss (g)	29.9 ± 18.9	27.0 ± 20.2	N.S.
Duration of surgery (min)	142.3 ± 30.8	133.7 ± 25.9	N.S.
Intraoperative neuromonitoring stimulation threshold (mA)	13.1 ± 5.4	18.5 ± 2.3	<0.001
Warning signals	20 (40.0%)	0 (0%)	<0.001
Nerve injury	5 (10.0%)	0 (0%)	<0.05

N.S.: not significant.

## Data Availability

Due to privacy and ethical constraints individual data cannot be disclosed.

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
