# Peer review of "Elimination of Lumbar Plexus Injury by Changing the Entry Point and Traction Direction of the Psoas Major Muscle in Transpsoas Lateral Lumbar Spine Surgery"

_medicina, 2023, doi:10.3390/medicina59040730_

Round 1
Reviewer 1 Report
Dear chief editor:
Thank you for sending this paper for reviewing. There were some comments in the following:
The study entitled “Elimination of lumbar plexus injury by changing the entry point and traction direction of the psoas major muscle in transpsoas lateral lumbar spine surgery” aims to investigate and compare neurological complications between the conventional lateral approach and a modified lateral approach at L4/5.
Comments and questions:
1. As the authors mentioned in the section of Materials and methods that they used T2-weighted MRI in the axial view to measure the relative position (RP) value. Please provide the MRI images to match up the illustrated figures already given in the manuscript.
2. The authors claimed in the section of Materials and methods that patients were classified into group X or group A based on the procedure they underwent. Please declare that, if patients were divided into the group randomly or by the other method.
In the section of Discussion, the authors wrote “In this modified splitting approach, it was proven that nerve injury did not completely occur by operating at an appropriate distance from the nerve.” Does it mean the nerve injury did not actually happen or others? And in the present study, the nerve injury did not occur (at least motor functions of hip flexors and knee extensors) by the modified splitting approach. Do you think the neuromonitoring is still essential (necessary) for this procedure to detect the distance to the nerve and prevent nerve damage.
Author Response
Dear chief editor:
Thank you for sending this paper for reviewing. There were some comments in the following:
The study entitled “Elimination of lumbar plexus injury by changing the entry point and traction direction of the psoas major muscle in transpsoas lateral lumbar spine surgery” aims to investigate and compare neurological complications between the conventional lateral approach and a modified lateral approach at L4/5.
Comments and questions:
- As the authors mentioned in the section of Materials and methods that they used T2-weighted MRI in the axial view to measure the relative position (RP) value. Please provide the MRI images to match up the illustrated figures already given in the manuscript.
We add the MRI images to the Figure 1 for understanding the RP value. The following sentence was added in the figure legend, “This patient is indicated for transpsoas lateral approach with an RP value of 65 on MRI evaluation.”
- The authors claimed in the section of Materials and methods that patients were classified into group X or group A based on the procedure they underwent. Please declare that, if patients were divided into the group randomly or by the other method.
We added the following explanation in Materials and methods.
“Since the procedure was changed in 2018, patients operated on before the change are in group X and patients operated on after the change are in group A.”
“All patients were followed for at least 2 years.”
In the section of Discussion, the authors wrote “In this modified splitting approach, it was proven that nerve injury did not completely occur by operating at an appropriate distance from the nerve.” Does it mean the nerve injury did not actually happen or others? And in the present study, the nerve injury did not occur (at least motor functions of hip flexors and knee extensors) by the modified splitting approach. Do you think the neuromonitoring is still essential (necessary) for this procedure to detect the distance to the nerve and prevent nerve damage.
We have changed the text to make it easier to understand as follows.
“This modified splitting approach has proven to cause no nerve damage at all by operating at the proper distance from the nerve.”
We believe that neuromonitoring is essential in measuring the distance to the nerve.
Reviewer 2 Report
This is a retrospective study to compare neurological complications between the conventional lateral approach and a modified lateral approach in 100 patients (50 patients in each group) with a lumbar degenerative disease requiring single-level intervertebral fusion at L4/5. The authors concluded that, when performing lumbar surgery using the lateral approach, lumbar plexus injury can be avoided by changing the surgical approach to the psoas muscle.
The following points seem worthy to mention:
1. Please explain the rational for using extreme lateral approach instead of conventional hemi laminectomy.
2. Table 1: Please delete Body height (m), and Body weight (kg) variables.
3. How long was the follow-up period of the patients?
Author Response
The following points seem worthy to mention:
- Please explain the rational for using extreme lateral approach instead of conventional hemi laminectomy.
We consider that fusion surgery is indicated for neuropathy at unstable segments. When foraminal stenosis is the cause of neurological symptoms, fusion surgery is also indicated because decompression of the spinal canal (e.g., laminectomy) does not improve the symptoms.
- Table 1: Please delete Body height (m), and Body weight (kg) variables.
Two items noted in Table 1 have been deleted.
- How long was the follow-up period of the patients?
Patients are followed up for a minimum of 2 years. We have added the follow-up period in Material and method.
Reviewer 3 Report
I would like to congratulate with the authors for a well and clearly written paper. The topic is very interesting, since XLIF is avoided by many spine surgeons for its sometimes-unacceptable neurological complications. The approach proposed by this study group is very interesting and should be validated by further studies, with a more unbiased design and a bigger sample. I would only suggest to better define the neurological complications that occurred in the x group (how many sensory, how many motory, how many transient, how many permanent).
Moreover, I would consider to modify Lines 189-191: This modified method that considered the relative positional relationship between the psoas muscle and the intervertebral disc and that adopted a modified point of entry to the psoas enabled us to perform lateral lumbar interbody fusion without lumbar plexus injury
Excellent work!
Author Response
I would like to congratulate with the authors for a well and clearly written paper. The topic is very interesting, since XLIF is avoided by many spine surgeons for its sometimes-unacceptable neurological complications. The approach proposed by this study group is very interesting and should be validated by further studies, with a more unbiased design and a bigger sample. I would only suggest to better define the neurological complications that occurred in the x group (how many sensory, how many motory, how many transient, how many permanent).
We have added a sentence in Result as follows,
“All patients in group X who had neuropathy developed both motor and sensory deficits, four improved within three months, but one was to remain permanently.“
Moreover, I would consider to modify Lines 189-191: This modified method that considered the relative positional relationship between the psoas muscle and the intervertebral disc and that adopted a modified point of entry to the psoas enabled us to perform lateral lumbar interbody fusion without lumbar plexus injury
We have modified the sentence.
“By considering the relative position of the psoas muscle and the intervertebral disc, and by changing the way the psoas muscle is entered and split, it is now possible to perform lateral lumbar intervertebral joint fusion without causing lumbar plexus injury.”
Excellent work!
Thank you for the wonderful feedback.
Round 2
Reviewer 1 Report
Dear chief editor:
Thank you for sending this paper for reviewing.
Comments:
A revised and improved version of the manuscript has been produced, which may be appropriate for publication.
Reviewer 2 Report
The authours have revised the article as requested.